# The Citizen Phage Library: Rapid Isolation of Phages for the Treatment of Antibiotic Resistant Infections in the UK

**DOI:** 10.3390/microorganisms12020253

**Published:** 2024-01-25

**Authors:** Julie Fletcher, Robyn Manley, Christian Fitch, Christina Bugert, Karen Moore, Audrey Farbos, Michelle Michelsen, Shayma Alathari, Nicola Senior, Alice Mills, Natalie Whitehead, James Soothill, Stephen Michell, Ben Temperton

**Affiliations:** 1Biosciences, Faculty of Health and Life Sciences, University of Exeter, Stocker Road, Exeter EX4 4QD, UKb.temperton@exeter.ac.uk (B.T.); 2Exeter Science Centre, Kaleider Studios, 45 Preston Street, Exeter EX1 1DF, UK; 3Microbiology, Virology and Infection Control, Great Ormond Street Hospital for Children NHS Trust, Great Ormond Street, London WC1N 3JH, UK

**Keywords:** antimicrobial resistance, phage therapy, phage cocktails, citizen science, named-patient phage therapy

## Abstract

Antimicrobial resistance poses one of the greatest threats to global health and there is an urgent need for new therapeutic options. Phages are viruses that infect and kill bacteria and phage therapy could provide a valuable tool for the treatment of multidrug-resistant infections. In this study, water samples collected by citizen scientists as part of the Citizen Phage Library (CPL) project, and wastewater samples from the Environment Agency yielded phages with activity against clinical strains *Klebsiella pneumoniae* BPRG1484 and *Enterobacter cloacae* BPRG1482. A total of 169 and 163 phages were found for *K. pneumoniae* and *E. cloacae*, respectively, within four days of receiving the strains. A third strain (*Escherichia coli* BPRG1486) demonstrated cross-reactivity with 42 *E. coli* phages already held in the CPL collection. Seed lots were prepared for four *K. pneumoniae* phages and a cocktail combining these phages was found to reduce melanisation in a *Galleria mellonella* infection model. The resources and protocols utilised by the Citizen Phage Library enabled the rapid isolation and characterisation of phages targeted against multiple strains. In the future, within a clearly defined regulatory framework, phage therapy could be made available on a named-patient basis within the UK.

## 1. Introduction

Antimicrobial resistance (AMR) has been declared by the World Health Organisation (WHO) as one of the top ten global public health threats facing humanity [1]. In 2019 there were an estimated 1.27 million deaths directly attributable to bacterial AMR, greater than the combined number of deaths from malaria and HIV/AIDS for that year [2]. Whilst there are innovative approaches to finding novel antibiotics [3,4] and progressive economic models to incentivise antibiotic development [5,6], developing new antibiotics remains scientifically and financially challenging. With the rapid rise of AMR there is an urgent need for new therapeutic options.

Bacteriophage (phage) therapy employs viruses that infect and kill bacterial pathogens [7]. Phage–bacteria interactions are highly specific and phage therapy has the advantage that, unlike antibiotics, there is no collateral damage to the host microbiome [8]. In Georgia, Russia and Poland, phage therapy has been widely used for decades [9,10] and has a proven track record for safety [11]. In western countries, the use of phages to treat bacterial infections fell by the wayside as the first antibiotics came into widespread use in the 1940s [12]. However, the rapid emergence of AMR has given rise to a resurgence in the use of phage therapy [13,14,15,16,17] and currently 15 phage therapy clinical trials listed on https://clinicaltrials.gov (accessed on 1 August 2023) [18] are actively recruiting.

One hurdle to achieving successful phage therapy is the development of bacterial resistance to the phage. Resistance to a particular phage can occur through a variety of mechanisms including altering surface phage receptors, degrading phage nucleic acids and triggering abortive infection systems [19,20]. An ideal broad-spectrum phage medicine would therefore take the form of a phage cocktail containing multiple different phages that not only collectively infect a large proportion of the circulating strains but also target different receptor binding proteins on host strains to minimise viable routes to evolved resistance [21]. To bring a new medicine to market in the UK it must have been prepared according to good manufacturing practice (GMP) standards and have undergone randomised controlled trials to assess safety and efficacy before an application can be made to license the product [22]. To achieve GMP production of phage therapy medicines in the UK will take time and require considerable investment. There is a strong case for an alternative, rapid turnaround, relatively low-cost pathway for phage therapy where patients with multidrug-resistant (MDR) infections have exhausted all other treatments.

Several countries are able to administer phage therapy via non-GMP pathways [14,15,23,24]. In the US for example, patients with life-threatening MDR infections can access phage therapy through the expanded access pathway [25] and in Belgium, a framework for phage therapy exists whereby magistral phage preparations prepared according to a standard monograph can be administered on a named-patient basis [22,26]. The STAMP study in Australia (standardised treatment and monitoring protocol to assess safety and tolerability of bacteriophage therapy for adult and paediatric patients) enables patients who have exhausted all other therapeutic options for control of their infection to access phage therapy [27]. Since 2019, there have been twelve cases of phage therapy in the UK using non-GMP prepared phage. Two cases have involved the treatment of *Mycobacterium* infections [14,28] (using phage discovered and prepared at the University of Pittsburgh, PA, USA) and in a more recent study a magistral phage preparation (phage discovered at the Eliava Institute, Georgia, phage prepared at the Queen Anne Military Hospital, Brussels, Belgium) was used to treat 10 patients with limb-threatening diabetic foot infections [29].

Phage therapy depends on the availability of a large and diverse bank of isolated phages, or the capacity to rapidly isolate suitable phages from the environment against a clinical pathogen. Numerous phage banks have been established globally and have provided phage for research and phage therapy [30]. One approach to building phage banks is through citizen scientist projects. In addition to educational benefits, engaging individuals in scientific research can increase scientific output. The SEA-PHAGES (Science Education Alliance–Phage Hunters Advancing Genomics and Evolutionary Science) program [31] has to date enrolled 1400 students and has banked 4471 fully sequenced Actinobacteriophages (https://phagesdb.org (accessed on 1 August 2023)). Phages from this program have been used to treat drug-resistant nontuberculous *Mycobacterium* infections, including infections in two patients in the UK as described above [14,28].

In 2020 the Citizen Phage Library (CPL, https://citizenphage.com (accessed on 1 December 2023)) was established in the UK to provide therapeutic phages to combat AMR through citizen science. Phage Hunters are invited to collect water samples from their local environment which are then sent to the CPL laboratory for the isolation of phages against WHO priority pathogens. The CPL currently holds over 1000 phages against pathogens such as *Acinetobacter baumannii*, *Pseudomonas aeruginosa*, *Escherichia coli*, *Staphylococcus aureus* and *Burkholderia* spp. In addition, the CPL holds nearly 600 water samples sent in by citizen scientists and several thousand wastewater samples provided by the Environment Agency from wastewater monitoring. These resources mean the CPL is well placed to screen clinical isolates for cross-reactivity with the banked phage collection and if no phages are available, the screening of water samples can be initiated immediately. To date, the CPL has responded successfully to four urgent need requests for phages made through the PhageDirectory (https://phage.directory/alerts (accessed on 1 December 2023)).

The study described here outlines the CPL response to a request from a UK hospital (via PhageDirectory) for phages against three MDR pathogens. The study highlights how protocols for high-throughput screening established by the CPL could be replicated to provide rapid access to phages for named-patient clinical use, supported by a clearly defined regulatory framework. The timeline for the discovery of phages against these strains is presented and the selection criteria for deciding which phages to take forward for the preparation of high titre purified seed lots are described.

## 2. Materials and Methods

Figure 1 represents an overview of the workflow for finding and selecting phages against incoming strains through to quality control checks of phage seed lots.

### 2.1. Maintenance and Sequencing of Bacterial Strains

Bacterial strains received from the hospital’s clinical microbiology laboratory were streaked onto LB agar and cryostocks were subsequently prepared using Pro-Lab Microbank vials (Fisher Scientific, Loughborough, UK). Three strains were received, a *Klebsiella pneumoniae* strain (assigned as BPRG 1484), an *Enterobacter cloacae* strain (assigned as BPRG 1482) and an *Escherichia coli* strain (assigned as BPRG 1486). All three strains were positive for IMP (active-on-imipenem) carbapenemase (Appendix A). Genomic DNA was extracted from the bacterial strains using the Nanobind CBB kit RT (PacBio, Menlo Park, CA, USA) according to the manufacturer’s instructions. Long-read sequencing of the bacterial genomes was carried out using a R9.4.1 (FLO-MIN 106) MinION flow cell (Oxford Nanopore Technologies (ONT), Oxford, UK). Library preparation was conducted using the Rapid Sequencing Kit (SQK-RAD004, ONT). Following priming of the flow cell, using the Flow Cell Priming Kit (EXP-FLP002, ONT), the library was loaded onto the flow cell. Data acquisition was achieved in real time using MinKNOW and basecalling was carried out using the Oxford Nanopore Guppy tool (v.6.0.1) using super-high accuracy models. Genomes were assembled using Trycycler (v0.5.3) and annotated with Prokka (v1.14.6).

### 2.2. Screening the Phage Collection against Bacterial Strains

Spot tests [32,33] were carried out as follows. LB broth supplemented with MgCl_2_ and CaCl_2_ (each at a final concentration of 10 mM, hereafter referred to as sLB) was inoculated with growth from the agar slope provided by the hospital laboratory and incubated at 37 °C with shaking (200 rpm) until the mid-logarithmic phase was reached (OD_600_ = 0.6). A 1 mL volume of the culture was added to 3 mL of cooled (55 °C) molten top agar, gently mixed and poured over the surface of pre-warmed bottom agar plates. Once set, phage lysates from the CPL collection were spotted (5 µL) onto the surface of the inoculated top agar and following overnight incubation at 37 °C, plates were examined for the presence of zones of lysis (ZOL). Top and bottom agar consisted of LB broth containing 0.65% *w*/*v* and 1% *w*/*v* bacteriological agar respectively. After cooling, MgCl_2_ and CaCl_2_ (each at a final concentration of 10 mM) were added to the top and bottom agar prior to use.

### 2.3. Screening Water Samples for New Phage against Bacterial Strains

Water samples sent in by citizen scientists and wastewater samples provided by the Environment Agency were screened for the presence of phage against the target *K. pneumoniae* and *E. cloacae* strains using a modified version of the high throughput method developed by Olsen et al. [33]. Citizen scientist water samples had been previously centrifuged at 10,000× *g* for 20 min (if required) to remove particulate matter, filtered through a 0.22 µm syringe filter (Sartorius, Goettingen, Germany) into sterile amber glass bottles and stored at 4 °C. Wastewater samples had been provided as either 1 mL aliquots in micro-centrifuge tubes stored at 4 °C or in frozen (−20 °C) 100 µL aliquots in microtitre trays. Wastewater samples were pooled (5–10 of the 1 mL aliquots or 30–40 of the 100 µL frozen aliquots), filtered (0.22 µm filter) into amber glass bottles and stored at 4 °C. For the screening assays, a 1 mL volume of each filtered water sample was added to a sterile 96 deep well plate (square well, V bottom, VWR, Radnor, PA, USA). To this was added 335 µL of 4.4 × concentrated LB broth, 60 µL of a 0.25 M MgCl_2_ plus 0.25 M CaCl_2_ stock solution and 70 µL of an overnight culture of the target strain. Plates were sealed with a SealPlate film (Merck, Darmstadt, Germany) and incubated overnight at 37 °C with shaking at 200 rpm (1st round enrichment). Following incubation, 200 µL from each well were transferred to a 96-well 0.45 µm MultiScreen_HTS_ HV sterile filter plate (Millipore, Merck, Darmstadt, Germany) placed on top of a 96-well microtitre plate (Grenier Bio-One, Kremsmünster, Austria). The double plate stack was centrifuged at 900× *g* for 4 min. A 5 µL volume of the filtrate was transferred to a new 96-well microtitre plate containing 95 µL sLB and 5 µL of an overnight culture of the target strain in each well. The plate was sealed as before and incubated overnight at 37 °C with shaking at 200 rpm (2nd round enrichment). Following incubation, 200 µL from each well were filtered as described previously. Filtrates from the 2nd round enrichment were spotted (5 µL) onto inoculated top agar overlays and following overnight incubation at 37 °C, plates were examined for the presence of ZOL. To purify the phage contained within the ZOL (which may contain more than one phage type), a core was taken from each clear zone using a sterile tip and transferred to 100 µL of SM buffer (50 mM Tris-HCl (pH 7.5), 0.1 M NaCl, 8 mM MgSO_4_). For axenic phage purification two methods were used and these are described in the supplementary material. For the wastewater samples, the volume of sample was limited and therefore for wastewater screening, the two target strains, *K. pneumoniae* and *E. cloacae*, were combined in the 1st round enrichment and 37.5 µL of an overnight culture of each strain were added to the deep well plate. The 2nd round enrichment was carried out on individual bacterial strains.

### 2.4. Phage DNA Extraction, Sequencing and Annotation

Phage infections were set up by adding 50 µL of purified phage and 500 µL of overnight culture to 20 mL of sLB (two tubes for each phage). Following overnight incubation at 37 °C, the infected cultures were centrifuged at 10,000× *g* for 20 min and the supernatant was filtered through a 0.22 µm syringe filter. A 30 mL volume of the resulting phage lysate was treated with 30 µL of DNase I and RNase A solution (10 mg/mL DNase I (Roche, Merck, Darmstadt, Germany) mixed with an equal volume of 20 mg/mL RNase A (Invitrogen, ThermoFisher Scientific, Waltham, MA, USA)) for 30 min at 37 °C to remove any remaining bacterial nucleic acid. Polyethylene glycol 8000 (PEG8000) and NaCl were added to the lysate to give final concentrations of 20% *w*/*v* and 2.5 M, respectively, and mixed gently by inversion until dissolved. The lysate was incubated at 4 °C overnight. Precipitated phages were concentrated via centrifugation at 10,000× *g* for 30 min and the phage pellet was re-suspended in 1 mL of SM buffer [34]. DNA was extracted from the phage using the Norgen phage DNA extraction kit (Norgen Biotek, Thorold, ON, Canada) following the manufacturer’s instructions. Phage DNA was quantified using the Qubit dsDNA broad range kit (Invitrogen, Waltham, MA, USA) and quality checked using the Genomic DNA ScreenTape assay on the Agilent 4200 TapeStation system. DNA libraries were prepared for short-read sequencing using NEBNext Ultra II FS Library Preparation and run on the Illumina Novaseq by the Exeter Sequencing Facility, to generate 2 × 150 bp paired-end reads. BBTools (v38.93) (https://sourceforge.net/projects/bbmap/ (accessed on 4 November 2022)) was used to remove optical duplicates, adapters and artefacts from the reads and error correction was carried out using Tadpole. Reads were mapped against host bacterial genomes to remove remaining residual host DNA and any DNA from prophages activated by lytic infection. Unmapped, corrected reads were assembled using Shovill (v1.1.0) using the following settings ‘--minlen 10,000 –mincov 20’. Assembled contigs were screened with CheckV (v1.01). Redundant overlapping ends were removed with apc (https://github.com/jfass/apc/ (accessed on 4 November 2022)). To evaluate whether new phages represented novel taxa, their closest relative at the nucleotide level was identified using the Mash distance as part of the INPHARED [35] pipeline. Reciprocal average nucleotide identity was then calculated between all closest relative genomes and all phage genomes isolated in this campaign using VIRIDIC [36], with novel species and genera assigned at 95% and 70% average nucleotide identity, as per the ICTV recommendations [37]. A phage was considered to represent a novel strain if its genome was less than 99% identical at the nucleotide level. Citizen scientists were invited to suggest names for their phage(s), which were then assigned as species names where appropriate. For further confirmation of the taxonomy of novel phages, nucleotide sequences from isolated phages were uploaded to the ViPTree Server (https://www.genome.jp/viptree (accessed on 9 January 2024), v 4.0) and aligned against dsDNA prokaryotic viruses in the Virus-Host DB to create a phylogenetic tree based on predicted proteome similarity [38]. The closest related phages in the database were assessed by alignment of phage genomes from the nearest shared inner branch of the generated tree. Full taxonomy information for each closest related phage was extracted using the NCBI Entrez Fetch command line tool and compared to nearest-neighbours identified by using INPHARED.

### 2.5. Phage Selection

Phages were selected for the preparation of seed lots from a panel of 20 *K. pneumoniae* phages and 22 *E. coli* phages according to the following criteria:Absence of temperate markers, antimicrobial resistance (AMR) genes and virulence factors. Phage genomes were annotated using Pharokka [39]. PhageAI (https://phage.ai/ (accessed on 8 November 2022)) [40] was used to predict the life cycle for each phage (virulent or temperate) and screening of phage genomes for AMR genes and virulence factors was carried out using Resistance Gene Identifier (RGI) [41] against the CARD database (https://card.mcmaster.ca/ (accessed on 8 November 2022)) and PhageLeads [42].Titre (PFU/mL) achieved on propagation on the hospital strains. Phage titres were determined using the small-drop plating assay [43]. A 1 mL volume of a mid-logarithmic culture (OD_600_ = 0.6) was added to 3 mL of cooled (55 °C) molten top agar, gently mixed and poured over the surface of a pre-warmed bottom agar plate. Ten-fold serial dilutions of the phage preparation were then spotted (5 µL) onto the inoculated agar overlay. Once the drops had dried, the plates were incubated at 37 °C and after overnight incubation, plaques were counted.Ability to suppress growth in liquid culture over a 24 h growth period. A mid-logarithmic culture of the bacterial host was diluted to an OD_600_ of 0.1 and 5 µL of this suspension was added to 95 µL of sLB broth in a microtitre plate. Phage lysates from the CPL collection (5 µL) were then added to each well as appropriate and the plate was incubated at 37 °C with periodic shaking in an Infinite^®^ 200 Pro plate reader (Tecan, Männedorf, Switzerland). Measurements of OD_600_ were recorded every 30 min for 24 h. For the control wells, 5 µL of sLB broth were added in -place of phage lysates.Genetic clustering of phage genomes. Heatmaps for the *K. pneumoniae* and *E. coli* phage genomes were generated using VIRIDIC [36] to determine the intergenomic similarities of phages in the selection panel. Phages were downselected to maximise diversity by selecting phages from different genomic clusters in order to increase the likelihood of generating a cocktail that targeted multiple receptor binding sites.

### 2.6. Preparation of Seed Lots

Phages were propagated in sequentially larger volumes as follows [44]. On the first day a 20 mL volume of sLB was inoculated with 50 µL of phage lysate and 500 µL of a mid-logarithmic (OD_600_ = 0.6) culture of the host organism. In this case, the *K. pneumoniae* BPRG 1484 strain was used as the propagation host. Following overnight incubation at 37 °C with shaking at 200 rpm, the infected culture was centrifuged and filtered using a 0.22 µm syringe filter to give a mini phage lysate. A larger volume (100 mL) of sLB was then inoculated with 250 µL of the mini phage lysate and 2.5 mL of an overnight culture and incubated and filtered as previously to give a midi phage lysate. Next, three flasks each containing 667 mL of sLB were inoculated with 1.67 mL of the midi phage lysate and 16.7 mL of an overnight culture. Following overnight incubation, the infected cultures were pooled and vacuum filtered through a 0.22 µm bottle-top filter unit (Fisher Scientific, Loughborough, UK) into a sterile glass bottle. The combined 2 L of filtered maxi phage lysate were transferred to a sterile 5 L flask and PEG8000 and NaCl were added to give final concentrations of 20% *w*/*v* and 2.5 M, respectively. A sterile stirrer bar was added and the lysate was stirred at 4 °C overnight. The lysate was then centrifuged in 200 mL volumes at 10,000× *g* for 30 min at 4 °C and the phage pellets were resuspended in 0.9% *w*/*v* sterile saline to give a total volume of 20 mL of precipitated phage suspension. The CsCl (UltraPure, optical grade, ThermoFisher Scientific, Waltham, MA, USA) gradient was prepared by layering 3 mL of successively dense solutions (1.2, 1.4, 1.5 and 1.6 g/mL prepared in autoclaved Milli-Q water) of CsCl into a 26.3 mL ultracentrifuge tube (Beckman Coulter, Brea, CA, USA) starting with the most dense solution at the bottom of the tube. Half of the precipitated phage suspension was then applied to the gradient. Following ultracentrifugation at 120,000× *g* for 3 h at 4 °C, the phage band was removed in a volume of ~2 mL by careful pipetting and dialysed extensively against sterile saline over 24 h. A Slide-A-Lyser dialysis cassette (Fisher Scientific, Loughborough, UK) with a 10 kDa MWCO was pre-wetted with sterile saline and ~2 mL of air were injected into the cassette via the injection port. A 1.5 mL volume of the phage suspension was injected into the dialysis cassette, which was then suspended in a sterile beaker containing 1 L of sterile saline and placed at 4 °C with stirring. The saline was replaced with fresh sterile saline at 18, 21 and 24 h. PFU/mL counts were carried out at each step of the seed lot preparation.

### 2.7. The In-House Quality Control of Seed Lots

Phage titre (PFU/mL) of the seed lots was determined as described above. The endotoxin concentration of the phage preparation was determined using the ToxinSensor chromogenic lyophilised amoebocyte lysate (LAL) endotoxin assay kit (GenScript, Pistcataway, NJ, USA). The assay was carried out according to the manufacturer’s instructions, using one-tenth volumes, in a pyrogen-free round-bottomed 96 well plate (Grenier Bio-One, Kremsmünster, Austria). Endotoxin standards over the range 0.0–1.0 EU/mL were prepared in the LAL reagent water provided. Tenfold serial dilutions of the seed lots were prepared in cell culture grade sodium chloride solution (0.9% *w*/*v*, Sigma-Aldrich, St. Louis, MO, USA). The standards and dilutions of the phage preparation were assayed for endotoxin concentration in triplicate. Short-read DNA sequencing of each seed lot was carried out as described previously to ensure that there had been no cross contamination with other phages and that there was no carry over of bacterial DNA from the propagating strain. The sterility of the phage seed lots was determined by spotting multiple 5 µL volumes of the seed lot onto LB agar, incubating overnight at 37 °C and then checking for the growth of microorganisms.

### 2.8. In Vivo Galleria Mellonella Infection Assay

The efficacy of *K. pneumoniae* phage cocktails (prepared from seed lots) was assessed using an in vivo *G. mellonella* infection model based on that described by Champion et al. [45]. Full experimental details can be found in the supplementary material. In order to determine the dose of *K. pneumoniae* bacterial cells required to induce signs of infection (death or melanisation), larvae were inoculated with increasing doses of *K. pneumoniae* cells (from 4.6 × 10^3^ to 4.6 × 10^6^ CFU) delivered in a 10 µL volume. Survival and melanisation were evaluated every 2 h for the first 8 h, then again at 20, 22 and 24 h. To assess if *K. pneumoniae* phage cocktails had any effect on the survival and melanisation of larvae infected with K. pneumoniae, larvae were inoculated with *K. pneumoniae* only (4.0 × 10^6^ CFU), phage cocktail only (2 × 10^7^ PFU) or *K. pneumoniae* and phage cocktail combined, as detailed in Appendix A. For larvae receiving both bacterial and phage injections, the bacterial inoculum was injected first, followed by a 30 min rest period prior to the phage cocktail injection. All larvae received two injections with control larvae receiving two saline injections. Once injections were complete, the larvae were transferred to 3D printed plastic trays with wells to separate and contain individual larvae (Appendix A). Trays were covered with fitted lids and incubated at 37 °C. To quantify melanisation, brightfield images of individual larvae were analysed using the software IMPACT2AMR (https://github.com/ashsmith88/IMPACT2AMR_galleria_imaging (accessed on 17 November 2022)), which identifies larvae within a boundary box and quantifies pixel brightness (inversely proportional to melanisation) within the boundary. Treatment groups (1–8) were distributed evenly across the plates to account for differences in light exposure across the plate that could affect melanisation scores (Appendix A). Survival and melanisation were recorded at 0, 14, 17, 21 and 24 h. Kaplan–Meier survival curves for the *G. mellonella* infection assays were plotted with the Survival Analysis Package v. 3.5–5 [46] using the survfit function and the survdiff function to test the difference between curves with a log rank test. To determine differences in the melanisation of larvae between treatment groups, the lme4 package v.1.1-35.1 [47] was used to run linear models, modelling melanisation as dependent on treatment, time (as a factor variable) and the interaction between treatment and time (melanisation~treatment*time).

## 3. Results

Initially, two strains were received from the clinical microbiology laboratory, a *K. pneumoniae* strain (BPRG 1484) and an *E. cloacae* strain (BPRG 1482), and work commenced on finding phages active against these two strains. Approximately two weeks later, an additional *E. coli* strain (BPRG 1486) was received and this was added to the schedule (Figure 2). Long-read whole genome sequencing was carried out for these strains and bacterial genome data, and associated reads are available on NCBI under BioProject PRJNA993854.

### 3.1. Cross-Reactivity of Target Strains with Phages Held in the CPL Collection Identified within One Day

Cross-reactions between *K. pneumoniae* and *E. coli* and phages already held in the CPL collection were identified within one day of receiving the bacterial strains. For the *K. pneumoniae* strain, 23 phages were screened yielding two ZOL. Sequencing data were only available for one of the phages (CPL00345). This phage was exclusively lytic and contained no virulence factors, toxins or AMR genes and hence was included as a potential candidate suitable for human use. In the case of the *E. coli* strain, 161 phages were screened and 42 ZOL were obtained (Figure 3). Full genome data were available for 23 of the 42 *E. coli* phages, with one phage (CPL00091) eliminated from being taken forward due to the presence of an integrase gene and two phages (CPL00134L and CPL00134S) found to have identical genomes.

### 3.2. New Phage Isolated against Target Strains within Four Days

A total of 169 ZOL were obtained from the water sample screening with *K. pneumoniae* (Figure 4a). Of these, nineteen clear ZOL were selected for purification and sequencing, with three of these originating from citizen scientist water samples and sixteen from wastewater samples. In the case of *E. cloacae*, 163 ZOL were obtained from the water sample screening (Figure 4b). Twenty clear ZOL were selected for purification and sequencing, with ten isolated from citizen scientist water samples and ten from wastewater samples.

### 3.3. Twenty-Nine Putatively Novel Phage Species Were Isolated

The newly isolated *K. pneumoniae* phage genomes were sequenced and characteristics of the phages together with the previously sequenced *E. coli* phages are presented in Table 1. Phylogenetic analysis revealed *K. pneumoniae* phages belonging to three different genera and *E. coli* phages belonging to nine different genera. Phages were considered to be novel species if they were <95% identical at the nucleotide level to previously known phages [37] and novel strains if they were <99% identical to previously known phages. In some cases, more than one isolated phage matched with the same closest known phage (Appendix A). Using these criteria, 16 novel species of *K. pneumoniae* phages were identified and 13 novel species of *E. coli* phages were identified, congruent with the greater number of phage genomes described for *E. coli* and hence greater phylogenetic coverage of *E. coli* phage genomes compared to K. *pneumoniae* phages, with 7075 and 1461 phage genomes currently registered with NCBI GenBank for *Escherichia* and *Klebsiella,* respectively. The taxonomic assignment of new phages to existing genera was consistent between ViPTree and INPHARED for all but two phages: CPL00221 and CPL00134L. Both of these phages were placed on their own branch in ViPTree (Appendix A) and shared <70% ANI from their nearest neighbour: *E. coli* phages CPL00221 (69.3% ANI with *Escherichia* phage AugustPiccard, MZ501051) and CPL00134L (47% ANI with *Escherichia* phage vB_EcoS_MM0, MK373793). These phages shared 41% ANI with each other, and <18% ANI with any other phage in this study. It is likely that these phages are the first representatives of a novel phage subfamily. Future sequencing efforts of phages from citizen samples will hopefully assist in finding additional members to support phylogenetic placement. The pooling of wastewater samples (16/20 *K. pneumoniae* phages were isolated from wastewater samples) may also lead to reduced diversity as the fittest phage in each pool will dominate. In contrast, 21/21 *E. coli* phages were isolated from individual environmental water samples from citizen scientists.

### 3.4. Titre Achieved on Propagation

The titre achieved on propagation in the hospital strains of *K. pneumoniae* (BPRG 1484) and *E. coli* (BPRG 1486) is given in Table 2. The *K. pneumoniae* phages produced titres ranging from 4.0 × 10^3^ to 4.4 × 10^11^ PFU/mL. These phages were isolated using the same *K. pneumoniae* strain (BPRG 1484), with the exception of CPL00345 which had been isolated using a strain of *K. pneumoniae* from a diabetic foot infection (DFI 20.19). CPL00345 gave the lowest titre of all of the *K. pneumoniae* phages when propagated in the *K. pneumoniae* BPRG 1484 strain (4.0 × 10^3^ PFU/mL). Of the 13 *E. coli* phages tested (which had been isolated initially on *E. coli* BW25113), 10 produced titres ranging from 1.5 × 10^6^ to 1.0 × 10^11^ PFU/mL when propagated in the hospital *E. coli strain* (BPRG 1486). Three of the phages (CPL00134L, CPL00221 and CPL00229) did not produce any plaques on the lowest dilution tested (×10^−2^). Thus, whilst the lysates of these three phages were able to produce a clear ZOL in the library collection screening assay (Figure 3) with *E. coli* BPRG 1486, they did not propagate well in this strain in liquid culture.

### 3.5. Phages Were Able to Suppress Bacterial Growth in Liquid Culture

*K. pneumoniae* and *E. coli* phages were assessed for their ability to suppress the growth of the target strains in liquid culture. Growth was monitored by measuring OD_600_ at 30 min time intervals for 24 h (Figure 5a,b) and the Virulence Index (*V_p_*) for each phage was calculated (Appendix A) [48]. In the case of *K. pneumoniae*, two phages were able to completely inhibit all growth over the 24 h growth period, whilst for *E. coli*, 12 phages (including T7) caused complete inhibition of all growth (Figure 5a,b). In some cases, a temporary increase in OD_600_ was observed followed by a drop and flattening of the growth curve. This is likely due to bacterial resistance to the phage followed by phage adaptation and the restoration of killing activity. For the *E. coli phages* CPL00134L, CPL00221 and CPL00229, very little or no inhibition of growth was observed (Figure 5b), and this correlated with their lack of ability to propagate in the *E. coli* BPRG 1486 strain (Table 2). Ability to suppress growth over a 24 h period was considered a strong indicator for suitability for seed lot preparation. If a phage was able to suppress bacterial growth over a 24 h period and yet had a lower titre than other phages in the selection panel, then this phage was still considered of interest; for example, the *K. pneumoniae* phage CPL00368 had a relatively low titre compared to other phages in the selection panel but performed well in liquid culture over 24 h (Figure 5a). Three *K. pneumoniae* phage lysate cocktails were tested for their ability to inhibit growth in liquid culture: cocktail 1—CPL00362 and CPL00368; cocktail 2—CPL00369 and CPL00379 and cocktail 3—CPL00362, CPL00368, CPL00369 and CPL00379. All three cocktails were able to inhibit the growth of *K. pneumoniae* over the 24 h period (Figure 6).

### 3.6. Phages against the Target Strains Exhibited Genomic Diversity and Clustering of Genomes into Distinct Groups

The *K. pneumoniae* phage genomes fell in to three distinct clusters whilst the *E. coli* phage were grouped into two large clusters, two small clusters of two phages and three phages which did not cluster with any other phages in the panel (Figure 7a,b).

### 3.7. K. pneumoniae Phages Were Selected for the Preparation of Seed Lots Based on Four Criteria

Phages were selected based on the absence of genes encoding unwanted characteristics, titre achieved on propagation (Table 2, ability to suppress growth in liquid culture over a 24 h growth period (Figure 5a,b) and genetic dissimilarity with other phages in the selection panel (Figure 7a,b). Four *K. pneumoniae* phages, namely CPL00362, CPL00368, CPL00369 and CPL00379, were chosen for the preparation of seed lots. The phage titre was determined at each step of the phage seed lot preparation (Appendix A). In-house quality control was carried out and the results are presented in Table 3 (and Appendix A). A phage dose of 10^9^ PFU is often cited as a standard dose [14] and hence this has been used here to calculate a theoretical number of doses based on the titre and volume (1.4 mL) of the seed lot. The number of theoretical doses can be doubled (given in brackets) as only half of the PEG8000/NaCl precipitated phage suspension was applied to the CsCl gradient. It should be noted that lower phage doses are also often described [49]. The annotated genomes of the four phages can be viewed in Appendix A.

### 3.8. Phage Cocktails Reduced the Melanisation of K. pneumoniae Infected G. mellonella Larvae

The first stage in the *G. mellonella* infection model experiments was to establish the dose of *K. pneumoniae* required to induce measurable signs of infection so that any rescue effect of the seed lot phage cocktails could be observed. A dose of 4.6 × 10^6^ CFU induced the melanisation of the larvae and caused larval death (Appendix A). In the phage cocktail experiment, larvae receiving a dose of 4.0 × 10^6^ CFU demonstrated significant melanisation over 24 h (Figure 8). Cocktail 3, which combined all four phages, was able to significantly reduce the melanisation of infected larvae and at 24 h, melanisation was significantly lower in this treatment group compared to infected larvae that had received saline in place of phage (linear model estimate ± s.e. = −5563.6 ± 2026.7, *p* = 0.006, Figure 8, Appendix A). When treated with two-phage cocktails, melanisation was reduced by 20–40% in the *K. pneumoniae* infected larvae compared to infected larvae that had received no phage (Figure 8), though the differences were not significant (Appendix A), with large standard error in melanisation observed even with 10 replicates per treatment. At 24 h, there was 100% survival in all treatment groups (Appendix A). After prolonged incubation (39 h), two deaths occurred in the *K. pneumoniae* infected larvae receiving saline in place of phage, but no deaths were observed at this point for infected larvae that had received phage cocktails (cocktails 1, 2 and 3). Uninfected larvae, that had received two saline injections or saline plus phage cocktail, demonstrated no melanisation over the course of the experiment.

## 4. Discussion

The O’Neill report commissioned by the UK Government in 2014 predicted that without urgent action, there would be an estimated 10 million deaths per year globally due to AMR [50]. Despite recent funding, policy and legislative initiatives, the pipeline for new antibacterials remains sparse [6]. Phage therapy, with a proven track record of safety, could make a vital contribution to combatting MDR infections [51]. Jones et al. [52] present a comprehensive vision for phage therapy in the UK where off-the-shelf, licensed, GMP-manufactured cocktails are available for use throughout the NHS. In cases where a cocktail is ineffective against a patient’s strain, a personalised non-licensed phage formulation could be prepared from a bank of ready prepared GMP-manufactured phages. To achieve this goal will take many years and considerable investment. Named-patient use of non-GMP manufactured phage preparations is permitted in several countries [14,22,26,27] and such a pathway could be lifesaving for patients in the UK who might benefit from phage therapy in the immediate future. Currently in the UK, unlicensed non-GMP phage can, at least in principle, be used on a named-patient basis where a clinician deems that all licensed alternatives have been exhausted, though navigation to successful use remains challenging. Thus, whilst in theory there are no regulatory barriers to the appropriate use of phages in the UK, a well-defined pathway and the resources to facilitate their use is needed. In this study a workflow is described for the preparation of bespoke phage seed lots against MDR pathogens received from a UK hospital.

Named-patient phage therapy needs to be swiftly available, adaptive, effective and safe. Four phage seed lots active against *K. pneumoniae* were ready for quality control assessment within 40 days with the clock running continuously. In the protocol described by Luong et al. [34], the time taken to complete all of the elements of the workflow added up to 20.5 days. As the CPL continues to expand its phage collection it will be possible to reduce the timescales further as the need for environmental screening for new phages against a patient strain diminishes. Timescales could be further reduced if the most useful phages were available as high-quality seed lots [26] and by increasing the phage–bacteria characterisation level whereby their host range, putative receptors, and activity under different environmental conditions is known [53]. The provision of phage(s) also needs to be flexible to accommodate the occurrence of bacterial resistance to the phage(s), an adaptive immune response in the patient or co-infection with other species.

In order to maximise the potential effectiveness of phage cocktails, phages were selected for cocktail formulation based on a number of criteria: phage titre on propagation, ability to suppress growth in liquid culture and genetic dissimilarity with other phages in the selection panel. Bacterial growth was monitored over a 24 h period to evaluate the efficacy of each phage against the target strain and to assess the development of bacterial resistance over time. Using this method, phages were selected that were able to maintain their activity over an extended period and phages where the rapid emergence of resistance was observed could be excluded. Furthermore, phages that produce a ZOL in a spot assay but do not perform well in liquid culture over 24 h, could be eliminated as in the case of the *E. coli* phages CPL00134L, CPL00221 and CPL00229. Haines et al. [54] found that, of the four methods tested, planktonic killing assays, similar to that described here, were the best method for determining phage virulence and for selecting phages for inclusion in cocktails. In the current study, phage lysates held in the CPL collection, were screened for their ability to suppress growth in liquid culture without a prior standardisation of their titre. This method provided a rapid screening tool to identify phages within the library with sustained activity against the target strain. Further analysis of the growth kinetics in liquid culture would be desirable, including one-step growth curves to determine phage adsorption rates and the optimal phage–bacteria ratio (multiplicity of infection, MOI) [32,53,55,56]. The phages in this study were isolated and their in vitro activity assessed, under physiological temperature and pH. When utilising phages for phage therapy, there will be multiple environmental factors operating in vivo which may affect the activity of a phage against its target host. These include temperature and pH [57], the presence of biological materials such as mucus [58,59], host immune responses and the interaction with the host microbiome [57,60]. Additionally, bacteria within an infection may be present as biofilms or may exhibit altered phenotypes which may hinder the adsorption of the phage to the bacterial surface [61]. The incorporation of the *G. mellonella* model into the workflow enabled confirmation of the in vivo efficacy of the four-phage *K. pneumoniae* cocktail in a simple insect model with an innate immune system. When assessing phages for their suitability for use in phage therapy, other routine screening assays such as biofilm assays and serum neutralisation assays could be included. One particularly important factor that may affect the activity of phages and phage cocktails in vivo is the presence of antibiotics that are to be administered alongside phage therapy. Antibiotics are often shown to have a synergistic effect when combined with phages though this is not always the case and antagonistic effects have been observed [62,63]. In future, synograms [64] could be included as part of the routine CPL workflow to assess in vitro phage activity in combination with the relevant antibiotic(s). Phage cocktails that maintain their efficacy in vivo would ideally be composed of phages with different target receptors on the bacterial host surface and differing susceptibilities to host bacterial defence mechanisms. By plotting VIRIDIC heatmaps of the phage genomes, phages were selected from different genome clusters to ensure maximum genetic diversity in the phage cocktail. In future it may be possible to use other approaches, such as data mining of host-range datasets, to improve cocktail design [65]. An important aspect of the efficacy of a medicine is evidence of its stability and resulting shelf-life. Phage activity may be affected by the storage buffer, its pH and the storage temperature [66]. Work is ongoing within the CPL laboratory to assess these criteria.

In addition to optimising phage cocktails for maximum potential efficacy, named-patient phage therapy needs to be safe. All phages were sequenced early on in the workflow and phages containing lysogenic genes and genes encoding virulence factors or antibiotic resistance were eliminated. Quality control criteria for phage seed lots include adequate titre, safe endotoxin levels, phage purity and sterility [22,50]. An endotoxin limit of 5 EU per kg of body weight per hour (or 350 EU for a 70 kg adult) for any parenteral route of administration other than intrathecal is specified by the U.S. Pharmacopeia [67]. In the current study, phages were purified by PEG8000/NaCl precipitation followed by ultracentrifugation and extensive dialysis against sterile saline. The endotoxin levels of the resulting seed stocks exceeded the required limit, and the method was time consuming and technically demanding. Further work is required to determine the best method for obtaining high titre phage preparations with the lowest achievable endotoxin levels. Hietala et al. [68] found that the most effective method of endotoxin removal was ultrafiltration (which concentrated phage and allowed buffer exchange, but did not reduce endotoxin-phage-ratio) followed by passing the preparation through an EndoTrap HD column. This method resulted in an endotoxin concentration of 0.085 EU/10^9^ PFU. Using cross-flow filtration, followed by LPS-affinity chromatography, Luong et al. [34] obtained an estimated 58–64,000 doses at 10^9^ PFU from 6 L batches with endotoxin levels ranging from 0.00025 to 0.07 EU per 10^9^ PFU dose. Purification steps such as endotoxin removal (Stage 4, Figure 1) would ideally be carried out in an environment with specified air quality and cleanliness. Whilst defined limits are specified for endotoxin levels, consideration should also be given to other toxins that may be produced by the propagation strain, such as the heat stable enterotoxins [69] and colibactin [70] produced by *K. pneumoniae* and *E. coli*. Purification protocols should in theory remove these alongside endotoxin, however evidence of the safety of a phage preparation could be confirmed with the use of cell viability assays [34]. The use of avirulent, low endotoxin propagation strains devoid of prophages where possible, would provide a more convenient route to achieving safe, non-toxic preparations [71]. Sequencing of the four *K. pneumoniae* seed lots was carried out and confirmed that the preparations were pure axenic phage with no evidence of cross contamination with other phages including prophages. Two of the seed lots contained significant amounts of host DNA, which could be reduced by introducing a second DNase/RNase treatment step after phage precipitation and prior to ultracentrifugation. Sterility was assessed by spreading an aliquot of each seed lot onto LB agar. This is perhaps adequate at this stage, as full sterility testing would be carried out by accredited laboratories if the seed lots were to progress to the preparation of phage active pharmaceutical ingredients (phage APIs) [26]. However, in future, in-house sterility testing will be assessed using bioburden testing on trypticase soya agar (Total Aerobic Microbial Count) and Sabouraud dextrose agar (Total Yeast and Mould Count) or by direct inoculation into tryptic soy broth and thioglycolate medium to ensure screening for the widest range of microorganisms. The workflow presented here ends with the in-house quality control of the purified seed lots. The continuation of this workflow to achieve high quality, non-GMP magistral standard APIs has been proposed in the final box in Figure 1, based on regimes utilised by Belgium and Australia [49,72].

This study has demonstrated that it was possible to identify phages targeted against multiple strains in a short time frame at a relatively low cost that are effective in vitro and in an in vivo *G. mellonella* infection model. Access to an extensive collection of water samples and biobanked phages expedited the screening process. In addition to the educational benefits, citizen science has proven highly productive in generating a diverse range of clinically suitable phages. An outline of how high quality non-GMP phages could be produced in the UK has been proposed and this, together with a clearly defined regulatory framework and adequate resources, could be used to provide phages for named-patient use in situations where all other therapeutic options have been exhausted. Further work is required to reduce the endotoxin levels of phage seed lots and to determine their stability over time. More detailed kinetic assays and phage–antibiotic synograms could be added to the workflow to further characterise the phage and enhance the selection process. The Citizen Phage Library will continue to engage with communities and expand its library of clinically suitable, fully characterised phages which can be provided for phage therapy globally. Recently, 175 fully characterised phages from the CPL were shared with PhageAustralia and a further 60 were shared with Canada to assist in their efforts, with additional shipments planned to support efforts in the US. The Citizen Phage Library will continue to work with the MHRA, clinicians and hospital pharmacies to establish a clear route to enable timely access to named-patient phage therapy in the UK.

## Figures and Tables

**Figure 1 microorganisms-12-00253-f001:**
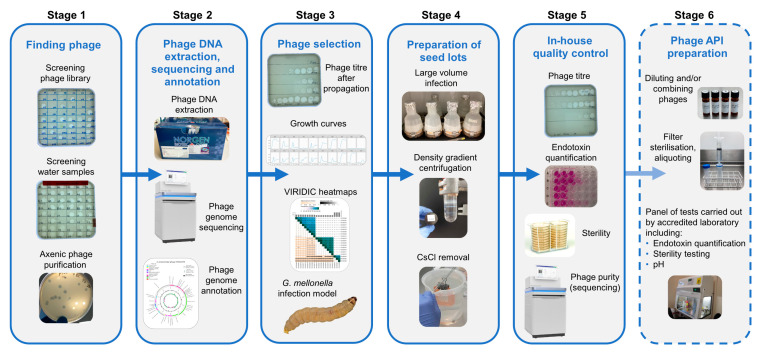
Overview of bacteriophage isolation, selection and preparation of seed lots. Stage1. Phages active against the target strain held within the currently available library are identified. New phages are isolated from citizen scientist water and Environment Agency wastewater samples and are purified to give axenic preparations. Stage 2. DNA is extracted from overnight infections, sequenced and annotated. Stage 3. Phages are selected based on a number of criteria: phage titre on propagation, ability to suppress growth in liquid culture and genetic dissimilarity with other phages in the selection panel. Stage 4. Phages deemed safe for human use are cultivated in 2 L volumes. Following centrifugation, the supernatant is 0.22 µm filtered to remove bacterial debris. Phages are precipitated from the filtrate using PEG8000 and NaCl. Following centrifugation, the precipitated phage is resuspended in 20 mL of sterile saline and applied to a CsCl gradient. The phage band obtained via ultracentrifugation is removed and dialysed extensively against sterile saline. Stage 5. In-house quality control on the seed lot to determine phage titre, endotoxin content, sterility and phage purity. Stage 6. For the proposed preparation of an active pharmaceutical ingredient (API), working in an environment with specified air quality and cleanliness, seed lots would be diluted (and combined if required), passed through a medical grade 0.22 µm filter and aliquoted into pharmaceutical grade vials. Quality control tests including endotoxin quantification, pH and sterility would be undertaken in MHRA accredited laboratories.

**Figure 2 microorganisms-12-00253-f002:**
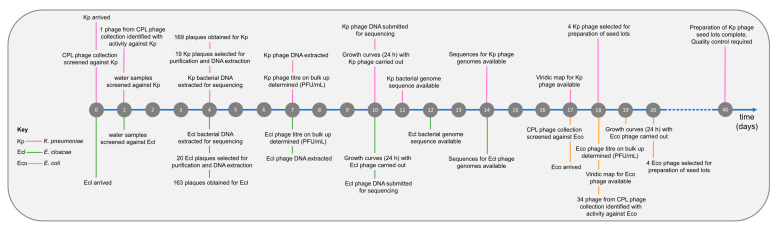
Timeline for phage search and preparation of seed lots against the hospital strains.

**Figure 3 microorganisms-12-00253-f003:**
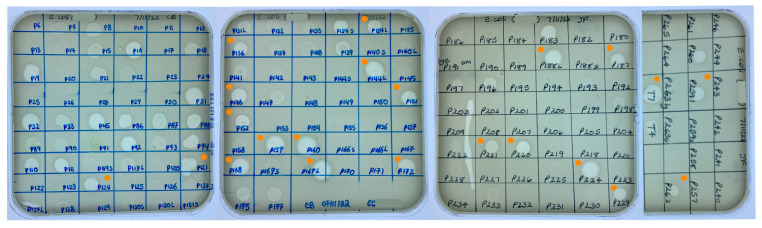
Screening of the CPL collection against *E. coli* BPRG 1486 determined by placing 5 µL spots of banked phage lysates on soft agar overlays of the strain. ● Phage lysates selected for further characterisation.

**Figure 4 microorganisms-12-00253-f004:**
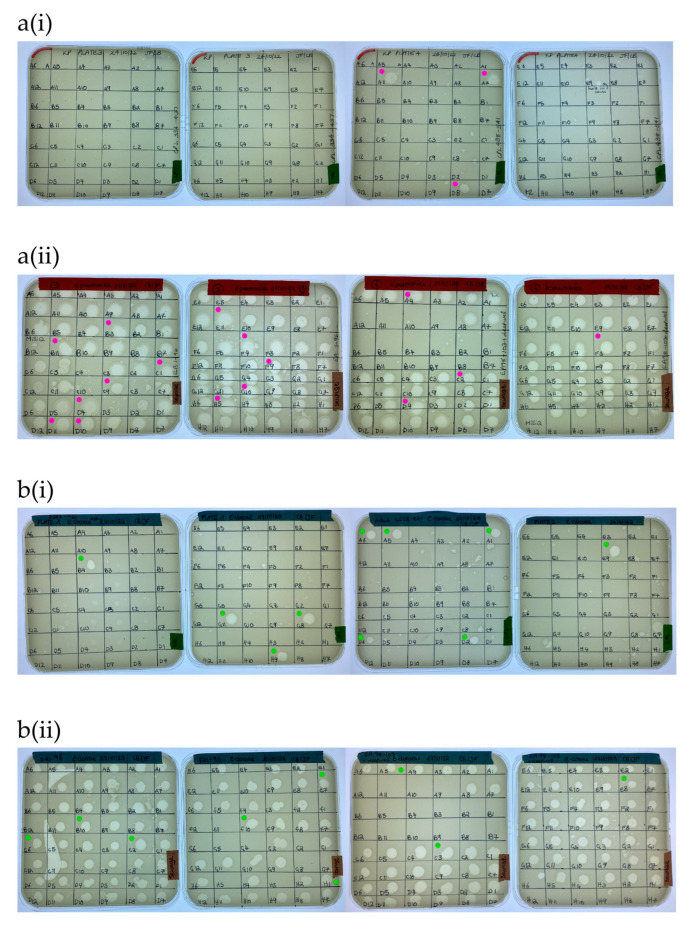
Screening of (**a**) *K. pneumoniae* BPRG 1484 with (**i**) citizen scientist water samples and (**ii**) sewage water samples. Cores taken for purification and sequencing from *K. pneumoniae* ZOL (●). Screening of (**b**) *E. cloacae* BPRG 1482 with (**i**) citizen scientist water samples and (**ii**) sewage water samples. Cores taken for purification and sequencing from *E. cloacae* ZOL (●). Activity was determined by placing 5 µL spots of 2nd round enrichment filtrates on inoculated soft agar overlays of the strains.

**Figure 5 microorganisms-12-00253-f005:**
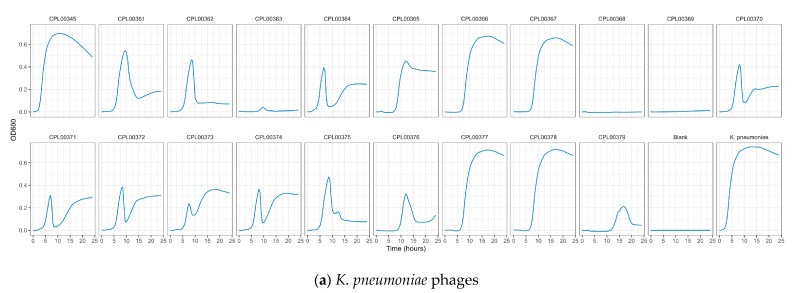
Ability of phages to suppress growth of (**a**) *K. pneumoniae* BPRG 1484 and (**b**) *E. coli* BPRG 1486 as determined by measuring OD_600_ at 30 min intervals over 24 h.

**Figure 6 microorganisms-12-00253-f006:**
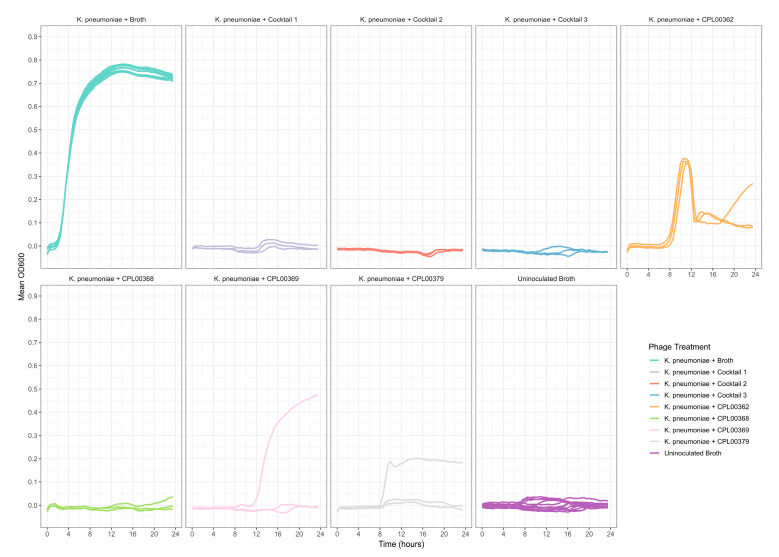
Ability of phage lysate cocktails to suppress the growth of *K. pneumoniae* BPRG 1484 as determined by measuring OD_600_ at 30 min intervals over 24 h. Cocktail 1—CPL00362 and CPL00368; cocktail 2—CPL00369 and CPL00379; cocktail 3—CPL00362, CPL00368, CPL00369 and CPL00379. Control wells received 5 µL of sLB in place of phage lysates.

**Figure 7 microorganisms-12-00253-f007:**
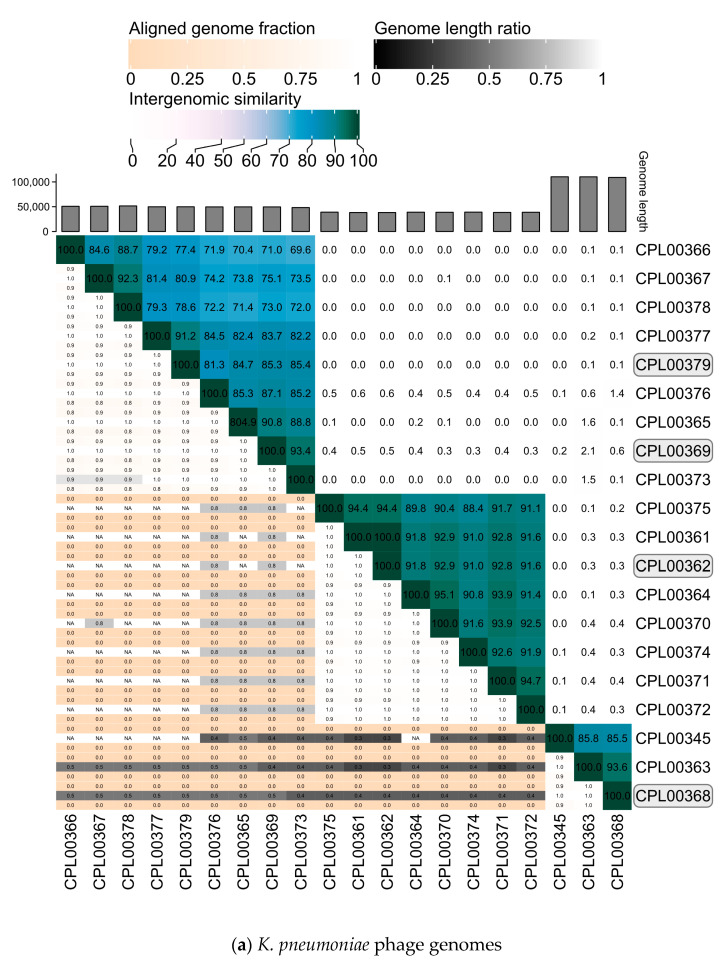
Heatmaps for (**a**) *K. pneumoniae* and (**b**) *E. coli* phage genomes plotted using VIRIDIC showing intergenomic similarities amongst the viral genomes. Highlighted phages indicate phages that were taken forward for the preparation of seed lots.

**Figure 8 microorganisms-12-00253-f008:**
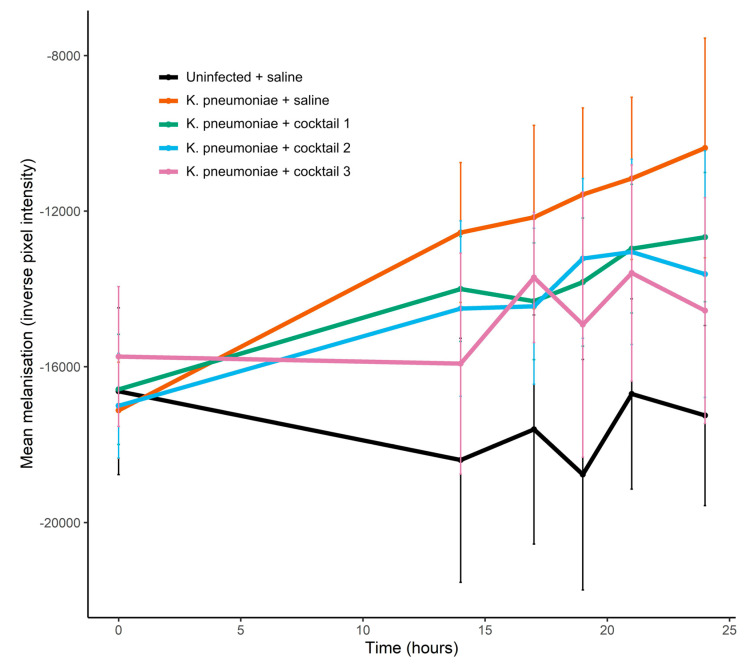
Effect of seed lot phage cocktails on melanisation of *K. pneumoniae* infected *G. mellonella* larvae. Each larvae received 2 injections: 10 µL of *K. pneumoniae* bacterial cells or 10 µL of saline, and 10 µL of phage cocktail or 10 µL of saline. *K. pneumoniae* dose = 4.0 × 10^6^ CFU delivered in a 10 µL volume. Phage cocktail dose = 2 × 10^7^ PFU in saline delivered in a 10 µL volume (with 1 × 10^7^ PFU for each phage in the two-phage cocktails and 5 × 10^6^ PFU for each phage in the four-phage cocktail). Cocktail 1—CPL00362 and CPL00368; cocktail 2—CPL00369 and CPL00379; cocktail 3—CPL00362, CPL00368, CPL00369 and CPL00379. Data are expressed as the mean ± s.e., *n* = 10.

**Table 1 microorganisms-12-00253-t001:** **(a)** Characteristics of the 20 phages active against the *K. pneumoniae* BPRG 1484 strain and their citizen scientist given names. **(b)** Characteristics of the 21 phages active against the *E. coli* BPRG 1486 strain and their citizen scientist given names.

	Phage	Sampling Location (what3words)	Outreach Event	Given Name	Family	Genus
**(a)**						
●	CPL00345	addicted.splints.evaded	Glastonbury	GlastoCabaret	*Demerecviridae*	*Sugarlandvirus*
	CPL00361	addicted.splints.evaded	Glastonbury	Identical to CPL00362	*Autographiviridae*	*Teetrevirus*
●	CPL00362	smart.dent.guard	Priorswood	Bobalons	*Autographiviridae*	*Teetrevirus*
●	CPL00363	agreed.trickling.evenly	Wooda Surgery		*Demerecviridae*	*Sugarlandvirus*
●	CPL00364	Environment Agency	Wastewater	PoeticCupcake	*Autographiviridae*	*Teetrevirus*
●	CPL00365	Environment Agency	Wastewater	SmellyBerry	*Drexlerviridae*	*Webervirus*
●	CPL00366	Environment Agency	Wastewater	RareGolfball	*Drexlerviridae*	*Webervirus*
●	CPL00367	Environment Agency	Wastewater	HelplessSquare	*Drexlerviridae*	*Webervirus*
●	CPL00368	Environment Agency	Wastewater	DevonBitter	*Demerecviridae*	*Sugarlandvirus*
●	CPL00369	Environment Agency	Wastewater		*Drexlerviridae*	*Webervirus*
●	CPL00370	Environment Agency	Wastewater	ViciousJeremy	*Autographiviridae*	*Teetrevirus*
●	CPL00371	Environment Agency	Wastewater	MegaDucksbill	*Autographiviridae*	*Teetrevirus*
●	CPL00372	Environment Agency	Wastewater	Bumbleweed	*Autographiviridae*	*Teetrevirus*
●	CPL00373	Environment Agency	Wastewater		*Drexlerviridae*	*Webervirus*
●	CPL00374	Environment Agency	Wastewater	Keithsmous	*Autographiviridae*	*Teetrevirus*
●	CPL00375	Environment Agency	Wastewater	Keithstache	*Autographiviridae*	*Teetrevirus*
●	CPL00376	Environment Agency	Wastewater	SlimeyKevin	*Drexlerviridae*	*Webervirus*
●	CPL00377	Environment Agency	Wastewater	MagicalPorter	*Drexlerviridae*	*Webervirus*
●	CPL00378	Environment Agency	Wastewater	AloofButler	*Drexlerviridae*	*Webervirus*
●	CPL00379	Environment Agency	Wastewater	StarXobjector	*Drexlerviridae*	*Webervirus*
**(b)**						
●	CPL00121	sweat.loudly.spends	Exeter Science Centre		*Drexlerviridae*	*Warwickvirus*
●	CPL00124	theme.deed.holly	Sidmouth Science Festival	Stokescottia	*Demerecviridae*	*Tequintavirus*
●	CPL00134L	plank.lobby.cars	University of Exeter	Phagiculus	*Drexlerviridae*	*Rogunavirinae* (subfamily)
●	CPL00136	during.laying.verge	University of Exeter		*Demerecviridae*	*Epseptimavirus*
●	CPL00144L	digit.issues.asks	Duke of Edinburgh	NorthRox	*Drexlerviridae*	*Warwickvirus*
●	CPL00146S	relate.rental.total	Duke of Edinburgh	SmurfNell	*Demerecviridae*	*Epseptimavirus*
●	CPL00151	just.most.smart	St James School		*Straboviridae*	*Krischvirus*
●	CPL00152	shovels.divide.owners	St James School	Aragogtheria	*Demerecviridae*	*Epseptimavirus*
●	CPL00159	wished.sage.avoid	Duke of Edinburgh	MikeNSara	*Drexlerviridae*	*Warwickvirus*
●	CPL00160	jumpy.movies.pure	Duke of Edinburgh		*Drexlerviridae*	*Warwickvirus*
●	CPL00168	pixel.strumming.inched	Bridgwater College Academy	MatMar	*Drexlerviridae*	*Warwickvirus*
●	CPL00169	users.limp.orders	Bridgwater College Academy	LinBro	*Ounavirinae* (subfamily)	*Felixounavirus*
●	CPL00169	users.limp.orders	Bridgwater College Academy		*Drexlerviridae*	*Warwickvirus*
●	CPL00172	endearing.tripling.called	Bridgwater College Academy	BubbaBully	*Drexlerviridae*	*Henuseptimavirus*
●	CPL00187	copper.jabs.circle	Yeo Valley Primary School		*Straboviridae*	*Krischvirus*
●	CPL00188L	copper.jabs.circle	Yeo Valley Primary School	WaterSpirit	*Drexlerviridae*	*Warwickvirus*
●	CPL00220	pirates.innocence.cheeks	University of Exeter	Baret	*Demerecviridae*	*Tequintavirus*
●	CPL00221	regard.coasters.cyber	University of Exeter	RobRod40	*Drexlerviridae*	*Christensenvirus*
	CPL00224	snack.skirt.rating	Exeter Science Centre	Identical to CPL00121	*Drexlerviridae*	*Warwickvirus*
●	CPL00229	softly.call.composers	Stoke Damerel Community College		*Demerecviridae*	*Tequintavirus*
●	CPL00259L	loser.words.smashes	Royal Society of Biology	McMelon		*Dhillonvirus*
●	CPL00262	loser.words.smashes	Royal Society of Biology		*Demerecviridae*	*Epseptimavirus*

(a) All phages were isolated using *K. pneumoniae* BPRG 1484 as the host, with the exception of CPL000345, which was originally isolated using *K. pneumoniae* DFI 20.19. ● represents a novel species, ● represents a novel strain. (b) All phages were isolated using *E. coli* BW25113 as the host. ● represents a novel species, ● represents a novel strain.

**Table 2 microorganisms-12-00253-t002:** Titre (PFU/mL) achieved for the *K. pneumoniae* phages and *E. coli* phages when propagated in the hospital strains.

Propagation Strain	Phage	Titre (PFU/mL)
*K. pneumoniae* BPRG 1484	CPL00345	4.0 × 10^3^
*K. pneumoniae* BPRG 1484	CPL00361	8.0 × 10^11^
*K. pneumoniae* BPRG 1484	CPL00362	8.0 × 10^8^
*K. pneumoniae* BPRG 1484	CPL00363	2.8 × 10^6^
*K. pneumoniae* BPRG 1484	CPL00364	2.0 × 10^7^
*K. pneumoniae* BPRG 1484	CPL00365	4.4 × 10^11^
*K. pneumoniae* BPRG 1484	CPL00366	4.4 × 10^7^
*K. pneumoniae* BPRG 1484	CPL00367	4.0 × 10^7^
*K. pneumoniae* BPRG 1484	CPL00368	1.6 × 10^5^
*K. pneumoniae* BPRG 1484	CPL00369	3.2 × 10^7^
*K. pneumoniae* BPRG 1484	CPL00370	8.0 × 10^8^
*K. pneumoniae* BPRG 1484	CPL00371	1.2 × 10^8^
*K. pneumoniae* BPRG 1484	CPL00372	2.8 × 10^8^
*K. pneumoniae* BPRG 1484	CPL00373	2.8 × 10^6^
*K. pneumoniae* BPRG 1484	CPL00374	4.0 × 10^6^
*K. pneumoniae* BPRG 1484	CPL00375	8.0 × 10^8^
*K. pneumoniae* BPRG 1484	CPL00376	1.2 × 10^6^
*K. pneumoniae* BPRG 1484	CPL00377	4.4 × 10^10^
*K. pneumoniae* BPRG 1484	CPL00378	4.0 × 10^9^
*K. pneumoniae* BPRG 1484	CPL00379	4.4 × 10^10^
*E. coli* BPRG 1486	CPL00134L	no ZOL (× 10^−2^ dilution)
*E. coli* BPRG 1486	CPL00144L	2.1 × 10^8^
*E. coli* BPRG 1486	CPL00151	3.2 × 10^6^
*E. coli* BPRG 1486	CPL00152	7.0 × 10^7^
*E. coli* BPRG 1486	CPL00159	1.3 × 10^11^
*E. coli* BPRG 1486	CPL00172	1.3 × 10^11^
*E. coli* BPRG 1486	CPL00188L	1.3 × 10^11^
*E. coli* BPRG 1486	CPL00220	8.0 × 10^6^
*E. coli* BPRG 1486	CPL00221	no ZOL (× 10^−2^ dilution)
*E. coli* BPRG 1486	CPL00224	1.9 × 10^8^
*E. coli* BPRG 1486	CPL00229	no ZOL (× 10^−2^ dilution)
*E. coli* BPRG 1486	CPL00259	1.5 × 10^6^
*E. coli* BPRG 1486	T7	1.3 × 10^8^

**Table 3 microorganisms-12-00253-t003:** In-house quality control of the *K. pneumoniae* seed lots.

Phage	PFU/mL	Theoretical Number of 10^9^ PFU Doses	Endotoxin (EU per 10^9^ PFU Dose)	% Host Reads	Evidence of Prophage
CPL00362	2 × 10^11^	280 (560)	8269	1.13	No
CPL00368	2 × 10^10^	28 (56)	17,150	0.12	No
CPL00369	5.2 × 10^11^	728 (1456)	784	15.92	No
CPL00379	4 × 10^11^	560 (1120)	582	25.3	No

## Data Availability

Bacterial and phage genome data, and associated reads, are available on NCBI under BioProject PRJNA993854. Data for the growth curve experiments, endotoxin assays and *G. mellonella* infection model assays are available on request.

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
