# Peer review of "The Citizen Phage Library: Rapid Isolation of Phages for the Treatment of Antibiotic Resistant Infections in the UK"

_microorganisms, 2024, doi:10.3390/microorganisms12020253_

Round 1
Reviewer 1 Report
Comments and Suggestions for Authors
Author Fletcher et al. describe "The Citizen Phage Library: Rapid isolation of phages for the treatment of antibiotic resistant infections in the UK".
Following are my comments to the editor for the decision:
1: Some sentences in the introduction section are without references.
2:2.2 Screening the phage collection against bacterial strains>>>provide references
3:2.4 Phage DNA extraction, sequencing and annotation>>>>provide a references
4:3.3 Nineteen novel phage strains were identified for K. pneumoniae and sixteen novel phage 381 strains were identified for E. coli, of which three were putative new species>>>>>>reduced this title
5: Figure 3 and 4.>>>would be more appropriate to label the positive result.
6:3.5 Phages were able to suppress bacterial growth in liquid culture over 24 h>>>>>no need to write 24 h here
7: Figure 5 and 6 can be presented in some nice way.
8: In figure 8, better to provide labelling inside the figure.
9: Why author determine the endotoxin content? There are several virulence characteristics of K. pneumoniae that should also need to be determined to be affected by phages.
10:Figure 1.>>>>label each panel with A, B, C, D, E and F. Then explain the meaning of these in the figure legends.
11: Enhance the resolution of Figure 2.
12: How about Tables 1a and b to put in the supplementary file?
13: Combine Table 2a and b
Comments on the Quality of English Language
Some minor spelling corrections are required
Author Response
Response to Reviewer 1
Dear Reviewer,
Thank you for your comments which we welcome and value. Please see below our response to your comments.
Author Fletcher et al. describe "The Citizen Phage Library: Rapid isolation of phages for the treatment of antibiotic resistant infections in the UK".
Following are my comments to the editor for the decision:
1: Some sentences in the introduction section are without references.
References have been added to support these sentences.
2:2.2 Screening the phage collection against bacterial strains>>>provide references
References for spot assays have now been added.
3:2.4 Phage DNA extraction, sequencing and annotation>>>>provide a references
A reference has been added to support these methods.
4:3.3 Nineteen novel phage strains were identified for K. pneumoniae and sixteen novel phage 381 strains were identified for E. coli, of which three were putative new species>>>>>>reduced this title
This title has been shortened.
5: Figure 3 and 4.>>>would be more appropriate to label the positive result.
Coloured dots have been added to Figure 3 to indicate which phage lysates were selected for further characterisation. Coloured dots have been added to Figure 4 to indicate which ZOL were selected for taking agar cores for purification and sequencing.
6:3.5 Phages were able to suppress bacterial growth in liquid culture over 24 h>>>>>no need to write 24 h here
This has been amended.
7: Figure 5 and 6 can be presented in some nice way.
We have considered presenting the data as a table of the virulence index for each phage however this does not show details such as a peak (initial resistance), followed by drop off (susceptible) at the beginning of the growth curve. Plotting them as separate graphs is the clearest way we have found to present the data.
8: In figure 8, better to provide labelling inside the figure.
The legend has been placed inside the figure to make better use of the space.
9: Why author determine the endotoxin content? There are several virulence characteristics of K. pneumoniae that should also need to be determined to be affected by phages.
We have added the following sentences: Whilst defined limits are specified for endotoxin levels, consideration should also be given to other toxins that may be produced by the propagation strain such as the heat stable enterotoxins [71] and colibactin [72] produced by K. pneumoniae and E. coli. Purification protocols should in theory remove these alongside endotoxin, however evidence of safety of the phage preparation could be confirmed by the use of cell viability assays [34]. The use of avirulent, low endotoxin propagation strains devoid of prophages where possible would provide a more convenient route to achieving safe, non-toxic preparations [73].
10:Figure 1.>>>>label each panel with A, B, C, D, E and F. Then explain the meaning of these in the figure legends.
Each panel has now been labelled with Stage 1, Stage 2, Stage 3 etc. and these steps are given in the figure legend.
11: Enhance the resolution of Figure 2.
We will work with the editors to check the resolution of Figure 2.
12: How about Tables 1a and b to put in the supplementary file?
Table 1a and 1b are less detailed versions of the full tables (which can be found in the supplementary materials). The authors felt it was important to include citizen scientist data in the heart of the paper. The tables include data such as where the water sample was taken (w3w), what type of outreach event (eg science festival, school workshop) and most importantly the phage names given by the citizen scientists.
13: Combine Table 2a and b
Table 2a and b have now been combined.
Comments on the Quality of English Language: Some minor spelling corrections are required.
Spelling and grammar has been carefully checked. United Kingdom English has been used throughout.
We thank you for your contribution and hope that you will now consider the manuscript acceptable for publication.
On behalf of all the authors and yours sincerely,
Julie Fletcher and Ben Temperton

Reviewer 2 Report
Comments and Suggestions for Authors
I confirm that I have reviewed a paper titled “The Citizen Phage Library: Rapid isolation of phages for the treatment of antibiotic resistant infections in the UK”. Given the global status, especially on antimicrobial resistance, there is an urgent need for an alternative solution to address this phenomenon. This study is so unique in the sense that it addresses one of the fundamental challenges faced by the world. Phage therapy has been applauded as the most suitable alternative solution to curb the emergence and spread of antimicrobial resistant pathogens and/or infections. In this paper, the authors have employed innovative methods to isolate and characterise the phages infecting E. coli and K. pneumoniae species.
While acknowledging the strength of this paper, there are some limitations, which are so critical, especially for the isolation and characterisation of phages intended for the bio-control purpose. Temperature, pH, one step growth curve and adsorption rate are very important parameters in phage work. The data obtained from these parameters may give an insight information regarding the stability of phages against those parameters (e.g. temperature, pH). In this study, the data the stability of phages against temperature and pH, including one step growth and phage adsorption rate were not presented. For this reason, it is difficult to conclude that the phages (some) isolated in this study could be considered as an ideal candidate for phage therapy. Thus, the data on phage stability against temperature, pH, growth curve, and adsorption rate must be included in this paper prior to its acceptance.
The authors used VIRIDIC tool for the taxonomic classification (genus and species level) of the phages and phage diversity. This tool classifies phages based on nucleotide-based intergenomic similarities amongst the phage genomes. I recommend that the authors must use other tools such MEGA, ViPtree etc to determine the diversity of the phages isolated in this study.
Author Response
Response to Reviewer 2
Dear Reviewer,
Thank you for your comments which we welcome and value.
The authors agree that temperature, pH, one step growth curves and phage adsorption rates are very important parameters in phage work. The current study was a real-life example of finding phages for a UK patient under time pressure and with limited resources. We acknowledge the limitations of the study and have included an additional discussion (see below) which will enable readers to see what we achieved, what the limitations were and how the work could be taken forward in the future.
Additionally, we have now included ViPTree analysis to illustrate the diversity of the phages and this data is presented in the Supplementary Material. The published guidance of the International Committee on Taxonomy of Viruses, and confirmed with their Chair (Dann Turner) by personal communication, is to use VIRIDIC for determining genomic novelty of phages, with established cutoffs of 95% and 70% for species and genera, respectively. Therefore, we have used this method to determine average nucleotide identity % of phages in this study to their nearest neighbours identified by INPHARED. For all but two phages, ViPTree and INPHARED were in agreement with genus assignment of their nearest neighbours, with the exceptions (CPL00221 and CPL00134L) sufficiently divergent to represent possible novel subfamilies. ANI% calculated by INPHARED using Mash scores was more conservative than by VIRIDIC. Consequently, thirty of the phages isolated in this study represent novel species using the VIRIDIC cutoffs. The text and tables of the manuscript have been updated accordingly, with methods and results adjusted to include the new VIPTree analysis.
We thank you for your contribution and hope that you will now consider the manuscript acceptable for publication.
On behalf of all the authors and yours sincerely,
Julie Fletcher and Ben Temperton
The paragraph given below has been added to the discussion:
In the current study, phage lysates held in the CPL collection, were screened for their ability to suppress growth in liquid culture without prior standardisation of their titre. This method provided a rapid screening tool to identify phages within the library with sustained activity against the target strain. Further analysis of the growth kinetics in liquid culture would be desirable including one step growth curves to determine phage adsorption rates and the optimal phage-bacteria ratio (multiplicity of infection, MOI)[32,52,54,55]. The phages in this study were isolated and their in vitro activity assessed, under physiological temperature and pH. When utilising phages for phage therapy, there will be multiple environmental factors operating in vivo which may affect the activity of a phage against its target host. These include temperature and pH [56] , presence of biological materials such as mucus [57,58], host immune responses and interaction with the host microbiome [56,59].Additionally, bacteria within an infection may be present as biofilms or may exhibit altered phenotypes which may hinder adsorption of the phage to the bacterial surface [60]. Incorporation of the G. mellonella model into the workflow enabled confirmation of in vivo efficacy of the 4-phage K. pneumoniae cocktail in a simple insect model with an innate immune system. When assessing phages for their suitability for use in phage therapy other routine screening assays such as biofilm assays and serum neutralisation assays could be included. One particularly important factor that may affect the activity of phages and phage cocktails in vivo is the presence of antibiotics that are to be administered alongside phage therapy. Antibiotics are often shown to have a synergistic effect when combined with phages though this is not always the case and antagonistic effects have been observed [61,62]. In future, synograms [63] could be included as part of the routine CPL workflow to assess in vitro phage activity in combination with the relevant antibiotic(s).
An important aspect of the efficacy of a medicine is evidence of its stability and resulting shelf-life. Phage activity may be affected by the storage buffer, its pH and the storage temperature [63] . Work is ongoing within the CPL laboratory to assess these criteria.
Reviewer 2 report
I confirm that I have reviewed a paper titled “The Citizen Phage Library: Rapid isolation of phages for the treatment of antibiotic resistant infections in the UK”. Given the global status, especially on antimicrobial resistance, there is an urgent need for an alternative solution to address this phenomenon. This study is so unique in the sense that it addresses one of the fundamental challenges faced by the world. Phage therapy has been applauded as the most suitable alternative solution to curb the emergence and spread of antimicrobial resistant pathogens and/or infections. In this paper, the authors have employed innovative methods to isolate and characterise the phages infecting E. coli and K. pneumoniae species.
While acknowledging the strength of this paper, there are some limitations, which are so critical, especially for the isolation and characterisation of phages intended for the bio-control purpose. Temperature, pH, one step growth curve and adsorption rate are very important parameters in phage work. The data obtained from these parameters may give an insight information regarding the stability of phages against those parameters (e.g. temperature, pH). In this study, the data the stability of phages against temperature and pH, including one step growth and phage adsorption rate were not presented. For this reason, it is difficult to conclude that the phages (some) isolated in this study could be considered as an ideal candidate for phage therapy. Thus, the data on phage stability against temperature, pH, growth curve, and adsorption rate must be included in this paper prior to its acceptance.
The authors used VIRIDIC tool for the taxonomic classification (genus and species level) of the phages and phage diversity. This tool classifies phages based on nucleotide-based intergenomic similarities amongst the phage genomes. I recommend that the authors must use other tools such MEGA, ViPtree etc to determine the diversity of the phages isolated in this study.

Round 2
Reviewer 2 Report
Comments and Suggestions for Authors
None.